# Chelation of Zinc with Biogenic Amino Acids: Description of Properties Using Balaban Index, Assessment of Biological Activity on *Spirostomum Ambiguum* Cellular Biosensor, Influence on Biofilms and Direct Antibacterial Action

**DOI:** 10.3390/ph15080979

**Published:** 2022-08-09

**Authors:** Alla V. Marukhlenko, Mariya A. Morozova, Arsène M. J. Mbarga, Nadezhda V. Antipova, Anton V. Syroeshkin, Irina V. Podoprigora, Tatiana V. Maksimova

**Affiliations:** 1Department of Pharmaceutical and Toxicological Chemistry, Peoples Friendship University of Russia (RUDN University), 6 Miklukho-Maklaya Street, 117198 Moscow, Russia; 2Department of Microbiology V.S.Kiktenko, Peoples Friendship University of Russia (RUDN University), 6 Miklukho-Maklaya Street, 117198 Moscow, Russia; 3Shemyakin-Ovchinnikov Institute of Bioorganic Chemistry, RAS, 16/10 Miklukho-Maklaya Street, 117997 Moscow, Russia

**Keywords:** coordination biochemistry, zinc, methionine, QSAR, Balaban index, biosensor *Spirostomum ambiguum*, laser light scattering, antibacterial properties

## Abstract

The complexation of biogenic molecules with metals is the widespread strategy in screening for new pharmaceuticals with improved therapeutic and physicochemical properties. This paper demonstrates the possibility of using simple QSAR modeling based on topological descriptors for chelates study. The presence of a relationship between the structure (J) and lipophilic properties (logP) of zinc complexes with amino acids, where two molecules coordinate the central atom through carboxyl oxygen and amino group nitrogen, and thus form a double ring structure, was predicted. Using a cellular biosensor model for Gly, Ala, Met, Val, Phe and their complexes Zn(AA)_2_, we experimentally confirmed the existence of a direct relationship between logP and biological activity (Ea). The results obtained using topological analysis, Spirotox method and microbiological testing allowed us to assume and prove that the chelate complex of zinc with methionine has the highest activity of inhibiting bacterial biofilms, while in aqueous solutions it does not reveal direct antibacterial effect.

## 1. Introduction

Many drugs possess modified pharmacological and toxicological properties when administered in the form of metallic complexes: complexation may affect the fate of chemical reaction, change the pKa and absorption profile, cause changes in distribution, plasma protein binding profile or allergenicity and potency and enhance the activity of enzymes [1,2,3,4]. The broad portfolio of new metal-based therapies progressing through clinical trials demonstrates the potential for new metal-containing compounds [5,6]. For example, zinc complexes of alpha-amino acids and their derivatives with a Zn(N_2_O_2_) coordination mode were found to have in vitro insulinomimetic activity [7]. Complexation of non-steroidal anti-inflammatory drugs with chromium and nickel provided masking of the inherent side effect of gastrointestinal hemorrhage and ulceration [8]. New synthetized complex of Zn(II) containing valine and dithio-carbamate proved to be active in the treatment of breast cancer [9]. The strategy of metal coordination is a potent tool to get numerous new antimicrobial agents [10]: the complexation of norfloxacin with Mn(II), Co(II) or Zn(II) modifies the antibacterial activity of the fluoroquinolone [11]; a series of cobalt, nickel, copper and zinc complexes of bidentate Schiff bases exhibited significantly enhanced anti-bacterial and antifungal activity [12]; the complexation of derivatives of sterically hindered o-diphenols and o-aminophenols with Cu(II), Co(II), Ni(II) and Zn(II) ions exhibited antiviral and antimicrobial activity with low toxicity [3]; new water-soluble zinc-glucose-citrate complex showed its enhanced antibacterial activity [13] and novel mixed iron (III) and zinc (II) complexes containing amino acids and isonitrosoacetophenone exhibited better antimicrobial activity than the ligands [14].

We also should not ignore the use of complexes for such purposes as the introduction in living organisms of some essential metal ions found to be deficient; this most often applies to zinc [3,15]. The human organism contains 2–3 g of zinc, about 0.1% of which should be replenished daily, but conservative estimates suggest that more than 25% of the world’s population is at risk of zinc deficiency [16,17,18]. It is known that Zn is transported as “free” Zn via a saturable, specific transport such as Zn importers from the ZIP family, but because this process is competitive with other metals (Cu, Ca, Fe) and dietary ligands (folic acid, phytates), the concentration of “free” Zn can be lowered considerably [17]. This is where complexing ligands come to the rescue, forming zinc-containing structures with metabolisms that are different from the metabolism of Zn in its inorganic sources: zinc gluconate is one of the most frequently mentioned supplementations for oral administration with relatively high Zn bioavailability [19]. In general, various biogenic compounds and drugs contain metal binding sites and form Zn complexes under aqueous physiological conditions [1,20,21].

The increased biological activity of metal chelates can be explained by the concept of cell permeability and chelation theory [22,23]. The chelation reduces the polarity of the metal ion due to partial sharing of its positive charge with donor ligands and delocalization of π electrons over the chelate ring [14,24].

Undoubtedly, various implications of chelate complexes in the biomedical field require studying of many aspects (nature, stability, toxicity, formation conditions) to understand how they act in biological processes [3]. As for zinc supplementation, the duality of zinc complexes action bears several challenges, since the question arises as to what comes first—the effective delivery of a deficient microelement or the antibiotic effect on the intestinal microflora [25,26].

We applied an original approach using topological indexes to predict possible existing relationships between the lipophilic properties of zinc chelate compounds and their structure, followed by the experimental verification of obtained results. The latter included assessment of complex biological activity by Arrhenius cell-death kinetics for *Spirostomum Ambiguum* biosensor, as well as the comparative study of chelate influence on biofilms and their direct antibacterial properties [27]. Choosing the ligand, we posed on the fact that a compound must enable ionophorism: α–Amino acids seemed to be nice candidates—water-soluble, they could act as perfect chelating ligands forming a five-member ring due to the presence of proton acceptor amino group (NH_2_) and the donor carboxylic acid group (COOH) in them [28]. Moreover, the conclusions of numerous studies indicated that zinc amino acid complexes provided more bioavailable zinc than zinc sulfate or zinc oxide [29,30]. At the same time, many publications highlighted the results indicating a direct antibacterial effect of these compounds on pathogenic microflora [20,31,32]. In this work we are trying to answer the question of whether it is correct to use topological indices to describe the coordination structures of zinc with amino acids (Zn(AA)_2_) and use them for prediction of the most active compound with potential antibacterial effect on pathogenic and opportunistic bacterial strains.

## 2. Results and Discussion

### 2.1. Prediction of Zn(AA)_2_ Properties Using Topological Descriptors

Topological descriptors have proven to be very simple and useful in predicting many molecular properties, but topological analysis traditionally belongs to organic chemistry [33]. Nobody has systematically used topological indices to estimate the properties of coordination compounds, considering molecular graph theory “too primitive” to cope with chelates [34]. Difficulties in applying topological indices stem from the fact that the molecular graph of a coordination compound is not as well defined as one of an organic compound. However, enclosed models for amino acid chelates rather well describe interactions between side chains and the central atom [34]. In our chelated compounds there are two types of bonds between Zn^2+^ and the amino acid (AA) ligand: the covalently polar bond Zn-O and the donor-acceptor bond Zn-N, together forming a double-core five-membered ring (Figure 1b).

As distance based topological indices can be successfully used for modeling, monitoring and estimating various physicochemical parameters as well as physiological activities, we selected the topological Balaban index (J) based on the distance matrix between the atoms of the compound as the steric descriptor [35].

The log P coefficient is well-known as one of the principal parameters for the estimation of lipophilicity of chemical compounds and determines their properties [36]. To express and evaluate the biological activity of the studying zinc complexes, the n-octanol/water partition coefficient, quantified as a logarithm of the octanol-water distribution constant (logP), was used.

The obtained J and logP values for the studied amino acids and their zinc chelate compounds are presented in Table 1.

It should be noted that the calculated logP value for the zinc sulfate molecule [37] turned out to be −4.16. Therefore, the received data confirms that upon chelation, the polarity of a metal ion is reduced due to the partial sharing of the positive charge with the donor groups of the ligand and as a consequence of overlap with the ligand orbitals [22].

For the studied samples of amino acids and their chelated compounds with zinc, a graph of the “structure”—“property” relationship was plotted, where the values of the topological descriptor (J) were deposited on the abscissa axis and the lipophilicity values (logP) were placed on the ordinate (Figure 2).

The discriminating ability of the Balaban index to the selected ten amino acids turned out to be rather low to distinguish them from each other by their lipophilicity values. However, the introduction of zinc cation into their structure with the formation of chelated compounds changed the picture of values distribution: a trend line close to linear dependence was found on the graph for Zn(AA)_2_, except for Zn(Lys)_2_ and Zn(Thr)_2_. The latter are out of the general indicated pattern—their polarity, achieved as a result of the complexation reaction, exceeded the control value of logP, corresponding to the zinc sulfate molecule.

According to Hansch’s theory, for binding to target molecules, the new produced substance should have medium lipophilicity—not to be highly hydrophilic and not to be highly lipophilic. This is since medicines should circulate in the bloodstream, and thus be soluble in water, but also penetrate cell membranes, due to solubility in fats [38]. The relationship between log P and some biological responses is often inverse parabolic, in which a maximum in the biological response occurred at some optimum log P value [39]. In the set of analyzed compounds, zinc methioninate met these criteria to a greater extent, with a value of logP = −2.44.

### 2.2. Experimental Assessment of Zn(AA)_2_ Biological Activity Using S. Ambiguum Cellular Biosensor

Eukaryotic unicellular organism *S. ambiguum* is a representative of protozoa, its biological reactions are not specialized and as an obligate oligotrophic organism it reacts on the presence of any toxicant dissolved in water by reducing its lifetime. It was shown that the death rate of *S. ambiguum* featured the Arrhenius type dependence on temperature, while the substances with high biological activity, characterized by low cell survival, correspond to low activation energy (^obs^Ea) values, consistent with the toxicity values (LD_50_) for higher hierarchical level organisms [40]. We used the *Spirotox* eukaryotic cell model to study the ligand–receptor interactions of Zn(AA)_2_ for the experimental evaluation of predictions obtained in the previous modeling. For the biological experiment, five amino acids were selected from the studied pool—phenylalanine, valine, methionine, alanine and glycine—the closest to the linear trend after complexation and having different lipophilicity values.

Experimentally obtained values of activation energy for amino acids were added on the graph «structure—property» to detect any correlation with the previously given values of molecules lipophilicity. In general, the pattern repeated, and Figure 3 shows clearly that phenylalanine is characterized by the highest activation energy (120 ± 6 kJ/mol, *n* = 5), while glycine has the minimum value in the studied sample (73.8 ± 15.5 kJ/mol, *n* = 5). Although the standard deviations of the results of a biological experiment do not allow us to regard the differences between amino acids as reliable, there is a relationship between the lipophilicity of an amino acid and its toxicity parameters—the lower the logP value, the greater the effect of the amino acid on the cellular biosensor. Based on the obtained data, it is possible to conclude that the predominant energy-dependent transport of amino acids into the organism of *S. Ambiguum* is implemented by the means of specific carriers [17].

As for the complexes, the curve, which was typical for amino acids on the graph “Balaban index—activation energy”, acquired a linear trend after ligand coordination with zinc, as well as in the case of lipophilicity (Figure 3b). Thus, the results of the biological experiment coincided with the prediction data, showing on one hand the strict relation between biological activity (^obs^Ea) and lipophilicity, while on the other hand indicating the ability of Balaban distance-based index to be used for the description of chelate structures. Finally results obtained using the QSAR-analysis and Spirotox-method made it possible to discover and prove that zinc methionine complex has higher biological activity than zinc glycinate, for which the antibiotic effect was previously proven [22,41,42,43].

### 2.3. Microbiological Screening: Comparative Study of the Effect of Zinc Amino Acid Complexes on Biofilm Formation in E. coli Culture

Bacteria of many species generate supracellular forms, biofilms, characterized by functional specialization of constituting cells laying out certain advantages for the colony [44]. For example, biofilms provide an environment for poor antibiotic penetration and horizontal transfer of virulence genes which favors the development of multidrug-resistant organisms [45]. The present study investigated the effect of Zn(AA)_2_ on multicellular behavior of *Escherichia coli* [46]. Polycellular forms of the studied bacteria cultured under standard conditions in liquid nutrient medium were detected by “Cluster-1”—MDL particle size laser diffraction analyzer [47]. This device, based on low-angle laser light scattering, provides data on cell distribution by size, which makes it achievable to characterize the morphofunctional state of the bacterial population [44].

The obtained results of numerical distribution of cells in *E. coli* BL 21culture for the studied samples of Zn(AA)_2_ and control are shown in Figure 4.

Particle size analysis allowed detecting cell forms ranging in size from 0.4 to 60 µm in all studied samples, with a twentyfold numerical increase in the control sample (Figure 4, insert). In the control sample the maximum of cells numerical distribution falls on a fraction of 20 µm, which indicates the existence of cellular agglomerates in the solution. Fractions of smaller sizes, corresponding to single cells, and fractions of larger sizes, corresponding to large agglomerates, are also present in the control solution.

In the studied solutions of zinc with valine, glycine and phenylalanine, fractions of particles with sizes of 4 µm and 19 µm are predominate (Figure 4). However, in solutions of zinc with methionine and alanine, a different picture was observed: with the numerical preservation of the fraction of single cells, 4 μm in size, a more notable effect on the formation of biofilms is observed—there is no pronounced maximum at 19 μm, but only a gentle shoulder is presented on the graph (Figure 4).

While analyzing the distribution of the total surface area of cell forms depending on their size in the range from 0.4 to 60 µm (Figure 5) we detected four size groups with maximum surface area values in the control solution—at 5, 20, 30 and 37 µm. The last three maxima indicate the formation and presence of various sized cellular associates in control [48]. A slightly visible maximum corresponding to the largest surface area group of cell forms with a size of 8.6 µm was found in the zinc methioninate solution. The absence of any maxima in the range from 20 to 60 µm indicates the inhibitory effect of zinc methioninate on the process of biofilm formation.

The obtained results indicate that all the studied zinc complexes with amino acids have antibacterial activity and are capable of inhibiting the formation of cellular associates, which is generally consistent with the literature data [20,49].

Thus, during the incubation of zinc chelate compounds with a bacterial culture, their influence on the direction of the equilibrium process of cellular transitions during the formation of polycellular forms was shown—the equilibrium shifted to the left with a predominance of unicellular bacteria (Figure 6). At the same time, the absence of dispersed particles smaller than 6 µm in the zinc methioninate solution shown in Figure 5 may indicate that zinc chelated compounds irreversibly shift the equilibrium with the formation of dead bacterial cells.

Based on the character of numerical distribution of *E. coli* cell particles in culture depending on their size, it can be concluded that among all the studied chelated compounds, zinc complexes with Met and Ala had the greatest effect on the retardation of bacterial growth of *E. coli* culture. Experimental values of biological activity and preliminary results of the antibiofilm activity forced us into studying the straight antibacterial effect of zinc methioninate on pathogenic test cultures.

### 2.4. Assessment of Antibacterial Activity of Zinc Methioninate Complex

Zinc methioninate complex was evaluated for its antibacterial activity against two standard Gram^+^ and Gram^−^ bacterial strains (*S. aureus ATCC 6538* and *E. coli ATCC 25922*) using the well diffusion method. As expected, 1024 µg/mL kanamycin solution as a positive control showed a large inhibition diameter and all negative controls did not show any inhibition zone against the tested bacteria. The test solution of zinc methioninate complex in concentration of 10.24 mmol/L also did not show any inhibition zone against standard Gram^+^ and Gram^−^ bacterial strains. Thus, the obtained results revealed that the water solution of zinc methioninate complex doesn’t hold straight antibacterial activity.

#### Determination of Minimum Inhibitory Concentrations (MIC) for Zinc Methioninate Complex

The MICs for zinc methioninate aqueous solution (10.24 mmol/L) were evaluated on 10 bacterial strains (four Gram positive, four Gram negative uropathogenic bacterial strains and two standard strains) using the microdilution method. The results of the experiment showed that the test solution of zinc methioninate complex in maximum studying concentration does not inhibit the growth of nine studied bacterial strains. MIC was only determined against *Staphylococcus aureus ATCC 6538* and turned out to be equal to 0.463 mg/mL.

### 2.5. Zn(Met)_2_ Impact on Biofilm Formation of Pathogenic and Conditionally Pathogenic Strains

Polycellular forms have been very well characterized over the past few decades because of their important role in the development of disease [50,51]. It has been proven that the formation of biofilms by bacteria can prevent and reduce the effectiveness of antibacterial therapy for infections. To prevent biofilm formation, scientists try to discover antimicrobial agents that affect the viability of bacteria in biofilms.

In the present study, we observed that zinc methioninate complex inhibited the biofilm formation in eight bacterial strains (three Gram positive bacterial strains, three Gram negative uropathogenic bacterial strains and two standard strains). *Conybacterium spp 1638* and *Citrobacter freundii 426* were excluded from this experiment because they did not demonstrate the ability to form biofilms and consequently any sensitivity to the test sample.

As shown in Figure 7, tested concentrations of Zn(Met)_2_ solution demonstrated percentages of inhibition varying from 9.71% to 100%. The percentage of biofilm inhibition on *Streptococcus agalactiae 3984, Acinetobacter baumannii 5841* and *Staphylococcus aureus ATCC 6538* at the concentration of 3.84 mmol/L was >80%; at the same time the percentage of biofilm inhibition on *Escherichia coli ATCC 25922* at the same concentration of zinc methioninate was 100%. Percentage of biofilm formation inhibition on *Enterococcus cloacae 6392* and *Escherichia coli M17* was less than 50% at the maximum test concentration.

The obtained results proved high antibiofilm activity (more than 70% of inhibition) against *Acinetobacter baumannii 5841*, *Streptococcus agalactiae 3984, Escherichia coli ATCC 25922* and *Staphylococcus aureus ATCC 6538* bacterial strains. The average antibiofilm activity (from 50% to 70% of inhibition) was detected on *Staphylococcus aureus 1449* and *Staphylococcus simulans 5882*, and poorly expressed antibiofilm activity (up to 50% of inhibition) was discovered on *Enterococcus cloacae 6392* and *Escherichia coli M17* bacterial strains. The high values of biofilm inhibition, determined for Gram^+^ and Gram^−^ standards, *Staphylococcus aureus ATCC 6538* and *Escherichia coli ATCC 25922* respectively, are because these strains did not possess any antibiotic resistance. *Escherichia coli M17* is a producent of monocomponent coli-containing probiotic and is used as lyophilized biomass of living non-pathogenic, non-toxic, active strains of *E. coli*. These bacteria are representatives of the normal microbiocenosis in humans and provide protection of their intestinal microflora [52,53,54]. Thus, the obtained results demonstrated that zinc methioninate does not affect the formation of *Escherichia coli M17* biofilms on the intestinal walls and probably does not disturb the normal microflora of the human body when it is systematically taken as a biologically active Zn-containing additive.

## 3. Materials and Methods

### 3.1. Chemicals and Media

Glycine anhydride (99.0%, Alfa Aesar, Kandel, Germany), L-Alanine (99.0%, Alfa Aesar, Kandel, Germany), L-Methionine (pure, pharma grade, AppliChem, Barcelona, Spain), L-Valine (99.0%, Alfa Aesar, Kandel, Germany), L-Phenylalanine (98.5%–101.0%, specified according to the requirements of USP, Acros Organics, Barcelona, Spain), zinc sulfate monohydrate (99.0%, Acros Organics, Barcelona, Spain) were used in experiments. All the media (BHIB: Brain Heart Infusion Broth; MHA: Muller Hinton Agar; SDB: Sabouraud Dextrose Broth) were procured from HiMedia™ Laboratories Pvt. Ltd., Thane West, India and all other reagents and chemicals (sodium hydroxide, ethanol) used were of analytical grade. For dilution, highly purified water was used, obtained using the Milli-Q^®^ purification system (Merck, Darmstadt, Germany).

### 3.2. Construction of Molecular Graphs and Balaban Index Calculation

Ten amino acids were selected as the objects of the study: Gly, Ala, Val, Leu, Ile, Met, Thr, Lys, Trp and Phe and their chelated compounds with Zn in the molar ratio Zn^2+^:amino acid = 1:2.

To predict the properties of chelated zinc compounds with amino acids a steric descriptor (Balaban index, *J*) and a thermodynamic descriptor (logarithm of the octanol-water distribution constant, logP) were used. Balaban indexes for amino acids were calculated using «ChemicPen» and «Chemical descript» software [55,56,57] and calculated manually for zinc chelated compounds. The value of the Balaban index was calculated as follows:
$$J=\frac{m}{\left(\mu +1\right)}\sum {\left(S_{i}\;S_{J}\right)}^{-0.5}=\frac{m}{\left(\mu +1\right)}\sum \frac{1}{\sqrt{S_{i}S_{j}}}$$
where ***S_i_S_j_***—the sum of the distances for the *i*-th and *j*-th vertices, summation is carried out for all adjacent vertices and ***μ***—the cyclomatic number of the graph that corresponds to the smallest number of edges whose removal leads to a graph without cycles.
***μ = m − n + q = m − n* + 1**(2)

***m***—number of edges (bonds)

***n***—number of vertices

***q***—the connectivity component of the graph, in Balaban’s studies *q* = 1.

To calculate the sum of distances for the *i*-th and *j*-th vertices (***S_i_S_j_***) the topological matrix was used (Table 2).

The following values are used to fill the topological matrix:-The values of the diagonal elements ***D_ii_*** of the topological distance matrix:
***D_ii_* = 1 − 6/*Z_i_***(3)
where ***Z_i_***—the number of all electrons in atom i
***D_ii_ (Zn-Zn) =* 1 − 6/30 = 0.8.**(4)-The values of the parameter ***P_ij_*** for the bonds of various atoms used in the calculation of non-diagonal elements of the topological distance matrix:
***P_ij_* = 36/*b_r_Z_i_Z_j_***(5)
where ***Z_i_*** and ***Z_j_***—the number of all electrons in atoms i and j, respectively, and ***b_r_***—the multiplicity of the bond between atoms i and j.
***P_ij_ (Zn-O) =* 36/1 × 30 × 8 = 0.15**(6)
***P_ij_ (Zn-N)* = 36/1 × 30 × 7 = 0.171**(7)

The same calculations were carried out for all chelated zinc compounds with amino acids. The values of lipophilicity for amino acids were found in the databases Pubchem (National Institutes of Health, Bethesda, MD, USA) and Drugbank (Canadian Institutes of Health Research, Ottawa, ON, Canada) and computed by Molinspiration v2018.10 [58] and ALOGPS 2.1 [59] software for chelated compounds.

### 3.3. Cellular Biosensor Spirostomum ambiguum for Testing the Biological Activity

#### 3.3.1. Test Solutions

The biological activity of zinc chelate compounds was studied based on the example of Zn complexes with five amino acids: Gly, Ala, Met, Val, Phe. Test solutions of Zn(AA)_2_ were obtained as a result of complexation reaction between ZnSO_4_*H_2_O and AA mixed in aqueous media. The equilibrium concentrations of the product were preliminary estimated at different metal–ligand ratios based on the data on the stability constants of zinc amino acid complexes [60]. In further experimental work the metal and a ligand were used in the stoichiometric ratio Zn:AA = 1:20 to shift the direction of the complexation reaction to the right side. Note that the adopted stoichiometry of the complex connection—Zn (AA) 2—is somewhat conditional and its exact establishment was not included in the tasks of this study. Obviously, in a twenty-fold excess of ligand in the solution, tris-complexes and even more complicated structures can be presented in the medium.

The selection criterion for the concentration of test solutions was the lifetime of the cell model—a time interval of 5–7 min was considered suitable for the observations under minimum temperature conditions. The experimentally determined concentration level of the complexing agent and ligand, which provides the necessary cell lifetime, was 2 mmol/L and 40 mmol/L for Zn^2+^ and AA, respectively. In the experiment with pure amino acids, the concentration was selected considering the solubility of the amino acid and based on the data that the value of ^obs^Ea for AA does not depend on the concentration in the interval 60–250 mM [40]. The pH level of obtained solutions was adjusted to a value of 6.80 with 8.5% NaOH solution. Trusted laboratory equipment was used in preparation of the studied solutions—analytical laboratory scales AS 220.x2 (Radwag, Radom, Poland), and basic laboratory pH- meter PB-11 (Sartorius, Göttingen, Germany).

#### 3.3.2. Cell Biosensor *Spirostomum ambiguum* and Research Technique

The test culture *Spirostomum ambiguum* (*S. ambiguum)* has been cultured in the laboratory to carry out studies of individual and combined biological activity of medicines. The protozoan ciliate *S. ambiguum* is characterized by tape-shaped, dorsal body shape (1–3 mm long), has a macronucleus clear-shaped and mouth apparatus up to the back third of the body. Compared to other objects of biological testing, *S. ambiguum* have several advantages, since they are eukaryotic organisms. Statistically reliable sensitivity to toxicants makes it possible to compare the response of protozoa with that of mammalians. Under favorable conditions in a low-mineralized environment, cells do not die for a period not exceeding their cell cycle (about 20 h). *S. ambiguum* is widely used as a test culture for toxicological and pharmacological studies. When it is introduced into an environment with chemical compounds, it dies within a time interval that is a function of concentration and temperature [61].

The experimental installation included a thermostatically controlled 5-hole plate (Lauda Alpha A6 thermostat, Göttingen, Germany) and an MBS-10 binocular. Low-power fluorescent daylight lamps (10 W) were used for additional lighting.

Experimental method: the experiment was carried out in a temperature range of 26–30 °C (in increments of 2 °C). One test infusoria and 250 µL of the test solution were introduced into each of the plate holes. Five measurements were carried out at each test temperature for each solution of the test sample. The cell lifetime was taken at the time interval from the moment of introducing infusoria to the solution to the moment of cell death. The cell death was determined as the moment of cell immobilization with no contractile reaction to mechanical irritation or as the rupture of the membrane with the release of the contents of the protoplasm outwards.

To calculate the observation activation energy of the ligand-induced cell transition (^obs^Ea) the results were represented in Arrhenius coordinates—the logarithm of the cell death rate [y= ln(1/t)] as a function of inverse temperature (x = 1/T) [62].

### 3.4. Comparative Study of the Zinc Amino Acid Complexes Effect on Biofilm Formation in E. coli Culture

#### 3.4.1. Bacterial Strain and Test Solutions

The ability of zinc chelated compounds to reduce the formation of biofilms in *E. coli BL 21* culture was studied on the example of five zinc complexes with Gly, Ala, Met, Val, Phe [63,64,65]. After filtration (0.22 µm, Millipore, Burlington, MA, USA), 0.25 mL of an aqueous solution of Zn(AA)_2_ (8 mmol/L) was introduced into 2 mL of LB culture medium, thus the concentration of Zn(AA)_2_ in the culture medium was 1 mmol/L. The culture medium was used as a control: 10 g of tryptone (Pronadisa, Madrid, Spain), 5 g of yeast extract (Helicon, Moscow, Russian Federation), 10 g of sodium chloride were dissolved in 800 mL of highly purified water and then diluted to 1 L with the same solvent. Control and test samples were prepared in 3 repetitions. The test and control solutions were inoculated with 10 µL of overnight culture (18 to 24 h at 37 °C and 100 rpm) of *E. coli BL 21* and then incubated at 37 °C for 12 h and 100 rpm. Before the measurements test and control solutions were diluted with culture medium in 1.5 and 12 times, respectively. The formation of biofilms in *E. coli* culture was evaluated by the number and size of dispersed phase particles using low-angle laser light scattering method.

#### 3.4.2. Particle Size Analysis with Laser Diffraction Method

Particle size analysis (numerical and surface area distribution of particles by their size) was implemented using “Cluster-1”—MDL—laser meter of dispersion (manufactured by the Institute of Colloid Chemistry and Chemistry of Water, Ukraine and RUDN University, Moscow, Russia) [47]. Laser light scattering occurs on optical inhomogeneities of the medium that appear when the refractive index changes. To characterize the “dispersibility” of the samples, the grain-size distributions by surface area and fractions number (“size spectra”) were subjected to analysis: laser light scattering/laser obscuration value (1–T), number (%), specific surface area (sm^2^/g).

### 3.5. Determination of Antimicrobial Activity of Zinc Methioninate Complex

#### 3.5.1. Bacterial Strains

To study antimicrobial activity in screening mode, we used 10 bacteria strains including 5 Gram positive, 5 Gram negative. Most of them were isolated from the urine of patients positively diagnosed with urinary tract infections: Gram^+^—*Staphylococus aureus 1449*, *Streptococcus agalactiae 3984, Conybacterium spp 1638* and *Staphylococcus simulans 5882*, and Gram^−^—*Escherichia coli M17*, *Citrobacter freundii 426*, *Enterococcus cloacae 6392* and *Acinetobacter baumannii 5841* [66]. The sensitivity of pathogenic bacteria strains to eight antibiotics was determined using modified Kirby–Bauer’s disk method and the multidrug resistance (MDR) index of each bacterium was calculated [67]. *Staphylococcus aureus ATCC 6538* and *Escherichia coli ATCC 25922* were used as standard Gram^+^ and Gram^−^ respectively. All strains were provided by the Department of Microbiology and Virology of the RUDN University.

#### 3.5.2. Test Solution of Zn(Met)_2_

By mixing 10.24 mmol/L solution ZnSO_4_*H_2_O and 204.80 mmol/L solution Met (in molar ratio 1:20) we obtained an aqueous solution of zinc methioninate complex in concentration of 10.24 mmol/L (3.7048 mg/mL). The criterion for selecting the maximum concentration of the tested zinc methioninate solution was the data on the limiting solubility of the amino acid in water (soluble in water—from 10 to 30 mL of water per 1.0 g of methionine) [68]. Distilled water, 204.80 mmol/L Met aqueous solution, 10.24 mmol/L ZnSO_4_*H_2_O aqueous solution and standard solution of kanamycin (1024 µg/mL) were used as a negative control. All samples were sterilized by microfiltration (0.22 μm). The obtained test solution was used to prepare the different concentrations used in the analytical process.

#### 3.5.3. Inoculum Preparation

Bacteria were cultured for 24 h at 37 °C in 10 mL of BHI broth. After incubation, the cells were collected by centrifugation (10,000× *g*, 4 °C, 10 min), washed twice with sterile saline and resuspended in 5 mL of sterile saline to achieve a concentration equivalent to McFarland 0.5 using DEN-1 McFarland Densitometer (Grant-bio, Cambridge, UK).

#### 3.5.4. Assessment of Antimicrobial Activity Using Well Diffusion Method

The well diffusion method was used to assess the antimicrobial activity of zinc methioninate solution. When screening for antimicrobial activity we used just standard Gram^+^ and Gram^−^ bacterial strains. Fifteen milliliters of sterile Muller Hinton Agar (for bacteria) was poured into petri dishes and 100 μL of each microorganism were spread. Wells with a capacity of 20 µL were drilled on the culture medium and 20 µL (at 10.24 mmol/L) of Zn(Met)_2_ and 20 µL of each negative control was added. All the trials were done in duplicate. After incubation at 37 °C for 24 h, the inhibition diameters were measured.

#### 3.5.5. Determination of Minimum Inhibitory Concentration (MIC)

MIC is the lowest concentration of antibacterial agent that completely inhibits the visible bacterial growth. One hundred microliters of broth (BHIB) was added to all the wells of sterile U-bottom 96-well microplates and zinc methioninate solution (at 10.24 mmol/L) was subjected to serial twofold dilution. Each column represented the test solution for single strain. The culture medium was used as a positive control. For each test well 10 μL of the respective inoculum was added (with turbidity equivalent to a 0.5 McFarland scale). Finally, the plates were covered and incubated at 37 °C for 24 h and after incubation, MIC was considered as the lowest concentration of the tested material that inhibited the visible growth of the bacteria.

### 3.6. Zinc Methioninate Complex Impact on Biofilm Formation

The antibiofilm activity was tested on 8 uropathogenic strains described earlier using microtiter dish for biofilm formation assay. Two-hundred microliters of the BHIB prepared with the solution of zinc methioninate complex to achieve different concentrations (0, 0.512, 1.240, 2.560, 3.840 and 5.120 mmol/L) was introduced in the sterile 96-well microtiter plate (4 wells for each test concentration). The BHIB free from test solution was used as a negative control. The BHIB free from bacterial culture was used as a positive control (Figure 8). The test wells were inoculated with 10 µL of overnight culture (18 to 24 h at 37 °C and 100 rpm) of bacteria. After 48 h of incubation at 37 °C, the medium was removed from the wells and replaced with 200 µL of 1% (*w*/*v*) crystal violet solution during 10–15 min. The wells were rinsed three times with distilled water prior to drying at 37 °C. The biofilm-bound crystal violet was solubilized in 200 µL of 99% ethanol.

The A_550_ was measured and used to calculate the inhibition percentage as in equation:(8)$$Inhibition\left(\%\right)=\frac{\left(Ao-A\;blank\right)-\left(Ax-A\;blank\right)}{\left(Ao-A\;blank\right)}\times 100$$
where *A_o_*—absorption in negative control, *A_blank_*—absorption in positive control, *A_x_*—absorption in test solution.

### 3.7. Data Processing

Data analysis was acquired from *n* ≥ 3 independent experiments and is presented as the mean ± standard deviation (SD). Data processing and plotting was performed using OriginPro 2017 (OriginLab, Northampton, MA, USA) software.

## 4. Conclusions

A variety of manifestations of the metal chelate complexes’ biological activity pose a difficult task for scientists to develop approaches to studying the properties and mechanisms of action of these structures. The results presented in this paper are yet another piece of the vast puzzle of coordination compound biochemistry, this time demonstrating the possibility of simple topological indexes (Balaban index) to be an effective tool for the description of chelate structure. The success of the obtained modeling result lies in the fact of its confirmation in a biological experiment using a cellular biosensor *S. ambiguum*. It illustrated how the process of chelation increases the lipophilic nature of the central metal atom and favors its permeation through the lipoid layer of the membrane.

Metal-based pharmaceuticals hold great promise, but their potential for toxicity limits their current use. Regarding zinc compounds, there has been controversy about their biological activity, casting doubt on continued use of Zn supplements if they reveal antibacterial effects. We demonstrated the advantage of using Zn(AA)_2_ complexes as Zn supplementation, which probably comes from their ability to utilize less saturable pathways for Zn uptake, and also protects from antagonistic factors in the diet. The results of microbiological experiments proved that zinc-containing supplements aimed at eliminating the deficiency of this element are safe in relation to the natural intestinal microflora, restraining the growth of pathogenic strains.

## Figures and Tables

**Figure 1 pharmaceuticals-15-00979-f001:**
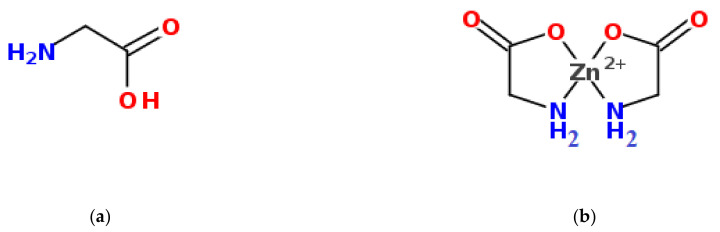
Chemical structures of glycine (**a**) and its zinc chelate compound—zinc glycinate (**b**).

**Figure 2 pharmaceuticals-15-00979-f002:**
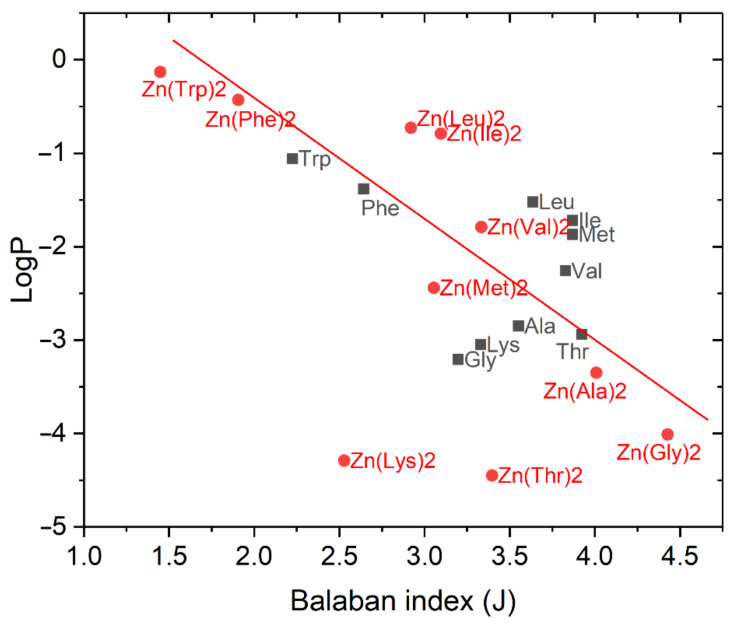
Dependence of lipophilicity on Balaban index for amino acids and their chelate compounds with Zn.

**Figure 3 pharmaceuticals-15-00979-f003:**
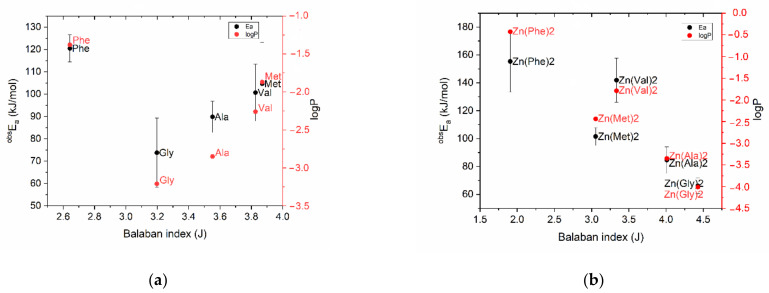
Dependence of biological activity (^obs^Ea) and lipophilicity on Balaban index for amino acids (**a**) and for their chelate compounds with Zn (**b**).

**Figure 4 pharmaceuticals-15-00979-f004:**
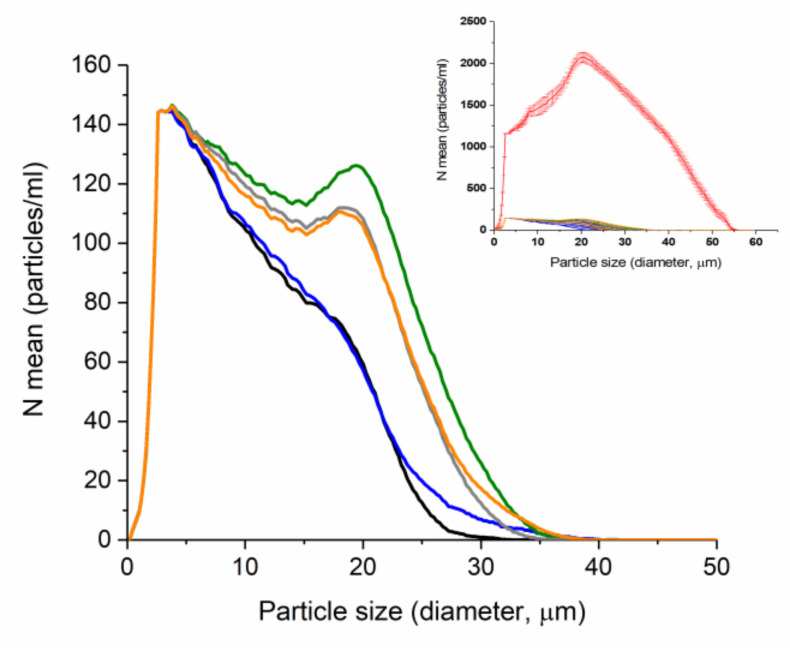
Comparison of the tested Zn(AA)2 complexes according to their inhibition influence on *E. coli* biofilm formation (*n* = 3), black—Zn(Ala)_2_, blue—Zn(Met)_2_, green—Zn(Val)_2_, grey—Zn(Gly)_2_, orange—Zn(Phe)_2_. Insert shows mean numerical particle size distribution for control (red) and test solutions of zinc amino acid complexes with SD (*n* = 3).

**Figure 5 pharmaceuticals-15-00979-f005:**
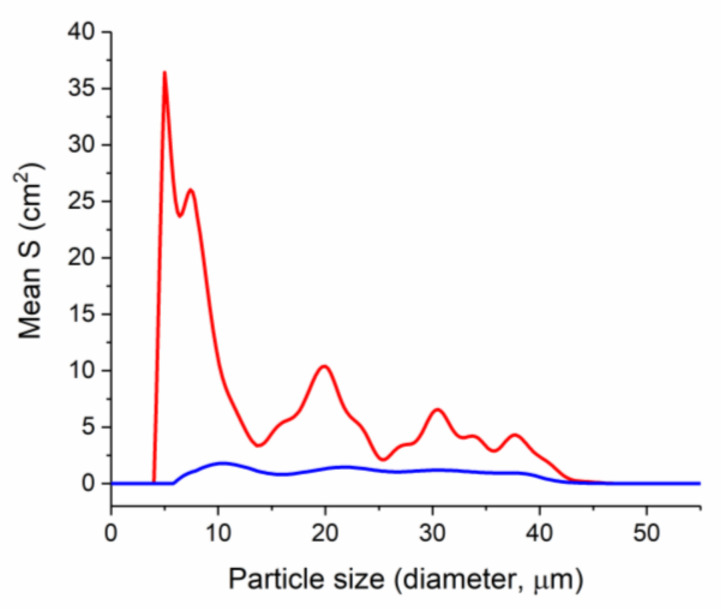
Comparison of the mean total particle surface area of the control and the test solution of zinc methioninate depending on the particle size (*n* = 3), red—control, blue—Zn(Met)_2_ solution.

**Figure 6 pharmaceuticals-15-00979-f006:**
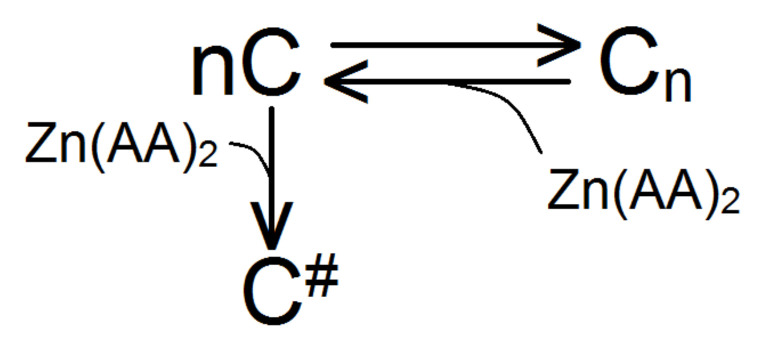
Scheme of cell transitions of bacterial culture in the presence of zinc chelate complex, C—single cell, C_n_—cellular associates of bacterial culture, C^#^—dead cells.

**Figure 7 pharmaceuticals-15-00979-f007:**
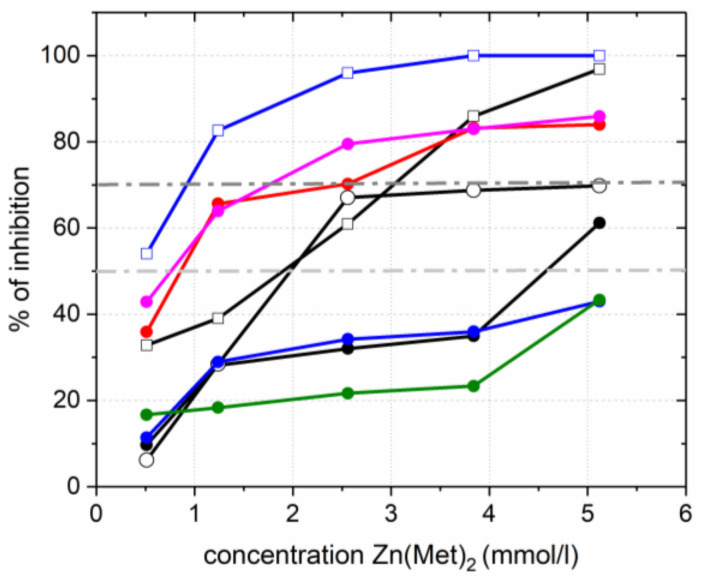
Antibiofilm activity of zinc methioninate complex, *Staphylococus aureus 1449*—circle black, *Staphylococcus aureus ATCC 6538*—unfilled square black, *Staphylococcus simulans 5882*—unfilled circle black, *Escherichia coli M17*—circle blue, *Escherichia coli ATCC 25922*—unfilled square blue, *Streptococcus agalactiae 3984*—circle red, *Enterococcus cloacae 6392*—circle green, *Acinetobacter baumannii 5841*—circle pink.

**Figure 8 pharmaceuticals-15-00979-f008:**
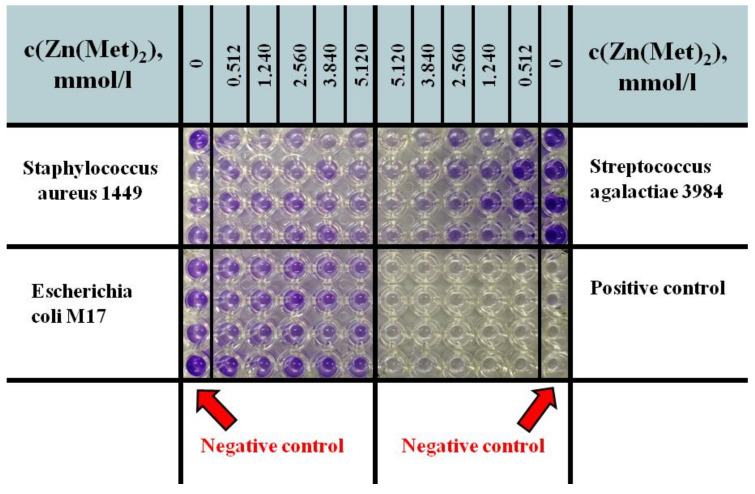
Scheme of antibiofilm activity determination on the example of *Staphylococcus aureus 1449, Escherichia coli M17* and *Streptococcus agalactiae 3984* bacterial strains: 96-well microtiter plate with 99% ethanol after the solubilization of biofilm-bound crystal violet.

**Table 1 pharmaceuticals-15-00979-t001:** The values of Balaban index (J) and logP for the studied amino acids and their zinc chelate compounds.

Name	AA		Zn(AA)_2_	
	J	logP	J	logP
Gly	3.1974	−3.21	4.4268	−4.01
Ala	3.5523	−2.85	4.0092	−3.35
Val	3.8267	−2.26	3.3345	−1.79
Leu	3.6362	−1.52	2.9205	−0.73
Ile	3.8679	−1.72	3.0964	−0.79
Met	3.8694	−1.87	3.0553	−2.44
Thr	3.9231	−2.94	3.3977	−4.45
Lys	3.3287	−3.05	2.5290	−4.29
Trp	2.2250	−1.06	1.4497	−0.13
Phe	2.6425	−1.38	1.9064	−0.43

**Table 2 pharmaceuticals-15-00979-t002:** Zinc glycinate’s topological matrix for Balaban index calculation.

N Atom i/j	1 N	2 C	3C	4 O	5 O	6 Zn	7 N	8 C	9 C	10 O	11 O	∑S_ij_
**1 N**	0.143	0.857	1.857	2.232	0.321	0.171	0.342	1.199	1.071	1.446	0.321	9.96
**2 C**	0.857	0	1	1.375	1.75	1.028	1.199	2.056	1.928	2.303	1.178	14.674
**3C**	1.857	1	0	0.375	0.75	0.9	1.071	1.928	1.8	2.175	1.05	12.906
**4 O**	2.232	1.375	0.375	0.25	1.125	1.275	1.446	2.303	2.175	2.55	1.425	16.531
**5 O**	0.321	1.75	0.75	1.125	0.25	0.15	0.321	1.178	1.05	1.425	0.3	8.62
**6 Zn**	0.171	1.028	0.9	1.275	0.15	0.8	0.171	1.028	0.9	1.275	0.15	7.848
**7 N**	0.342	1.199	1.071	1.446	0.321	0.171	0.143	0.857	1.857	2.232	0.321	9.96
**8 C**	1.199	2.056	1.928	2.303	1.178	1.028	0.857	0	1	1.375	1.75	14.674
**9 C**	1.071	1.928	1.8	2.175	1.05	0.9	1.857	1	0	0.375	0.75	12.906
**10 O**	1.446	2.303	2.175	2.55	1.425	1.275	2.232	1.375	0.375	0.25	1.125	16.531
**11 O**	0.321	1.178	1.05	1.425	0.3	0.15	0.321	1.75	0.75	1.125	0.25	8.62

## Data Availability

Data is contained within the article.

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
