# Peer review of "Chelation of Zinc with Biogenic Amino Acids: Description of Properties Using Balaban Index, Assessment of Biological Activity on Spirostomum Ambiguum Cellular Biosensor, Influence on Biofilms and Direct Antibacterial Action"

_pharmaceuticals, 2022, doi:10.3390/ph15080979_

Round 1

Reviewer 1 Report

The manuscript title “Chelation of zinc with biogenic amino acids: description of properties using Balaban index, assessment of biological activ-ity on Spirostomum Ambiguum cellular biosensor, influence on biofilms and direct antibacterial action” is well written and have scientific value. Minor comments for authors are as follows: 

Reviewer Comments:

1-      Please do italic all the scientific names in the manuscript.

2-      Please rewrite this sentence and put the reference in the end of sentence “The authors of [9] synthetized and described a new complex of Zn(II) containing valine and dithiocarbamate that can be used in the treatment of breast cancer.”

3-      I suggest to authors to remove the referenced statements from conclusion and only write the conclusion of your study/results. You can move the references in the discussion section.

Author Response

Dear reviewer, all needed corrections were done:

1) All scientific names in manuscript were changed to italic

2) The indicated sentence was changed: New synthetized complex of Zn(II) containing valine and dithio-carbamate proved to be active in the treatment of breast cance [9].

3) The links presented in conclusion were given more to confirm our own ideas, in some ways they echo with the introduction. Without regret, we will get rid of them in the indicated section of the article.

Reviewer 2 Report

This is an interesting approach for the evaluation of the biological activity of various zinc(II) complexes. This may support publication but some corrections are required.

1. The structure in Figure 1b must be corrected. Two protons on the amino groups.

2. 1:20 metal to ligand ratio and pH 6.8 was selected to prepare the Zn(AA)2 complexes. Under this conditions the existence of mono- Zn(AA) and tris Zn(AA)3 complexes is also possible. Clear evidence must be presented for the predominance of the bis complexes.

3,The length of the manuscript is too long as compared to its new scientific content. Significant reduction (especially in the Introduction) is necessary.

Author Response

Dear reviewer, we introduced demanded corrections:

1) Fig. 1b - corrected. Please, see the attachment. 

2) The establishment of the exact structure of the complex was not included in our plans. Of course, in aqueous solution under such conditions, the formation of both mono- and tris zinc aminocomplex is possible, the coordination of water molecules and sulfate ions is also not excluded. In addition, the literature describes polymer structure characterizing the crystalline metionine of zinc. For us, the excess of ligand and pH was important for a complete shift in the balance of the reaction and preventing the precipitation of zinc hydroxide, in addition, we wanted to limit the biological effect of the acidity of the medium. In other words, the experiment required a neutral environment, however, at a pH of 6-7 and a molar ratio of metal-ligand 1 to 2, a precipitate of zinc hydroxide fell, the stabilizing action of the ligand was not enough in such a concentration. In addition, we conducted a number of experiments on S. Ambiguum with a molar ratio of zinc and amino acids 1 to 2 - then for all amino acids the values ​​of the activation energy turned out to be almost the same, which probably indicated the toxic effect of free zinc, and not the influence of the complex.

The corresponding comments added materials and methods to the section. Please, see the attachment. 

3) The introduction and some other parts of the article were reduced. Please, see the attachment. 
